# Different Physiochemical Properties of Novel Fibre Sources in the Diet of Weaned Pigs Influence Animal Performance, Nutrient Digestibility, and Caecal Fermentation

**DOI:** 10.3390/ani14172612

**Published:** 2024-09-08

**Authors:** Agnieszka Rybicka, Pedro Medel, Emilio Gómez, María Dolores Carro, Javier García

**Affiliations:** 1Departamento de Producción Agraria, Universidad Politécnica de Madrid, 28040 Madrid, Spain; belaraniel@gmail.com (A.R.); mariadolores.carro@upm.es (M.D.C.); 2Innovabiotics, S.L., 28906 Madrid, Spain; pmedel@innovabiotics.eu; 3Centro de Pruebas de Porcino, ITACyL, Hontalbilla, 40353 Segovia, Spain; gomizqem@itacyl.es

**Keywords:** digestibility, fibre sources, fermentability, hydration capacity, performance, piglets

## Abstract

**Simple Summary:**

Dietary fibre was once considered to be a negative factor for monogastric animals due to its potential adverse effects on digestibility and performance. Recently, certain types of fibre have been proposed as an effective nutritional approach to maintaining piglets’ gastrointestinal health after weaning. However, the relationships between the complex fibre properties of different sources and their physiological effects remain unclear. This research had a dual purpose. Firstly, to evaluate the use of novel fibre sources in piglet feed, such as almond shell, olive kernel, and nutshell, which are very finely ground fibre sources typically unsuitable as dietary ingredients due to their hardness and particle size. Secondly, to assess the impact of their physicochemical properties, such as fermentability and hydration properties, on piglet performance compared with lignocellulose, the most commonly used fibre source. Overall, these results highlight the importance of the physicochemical characteristics of fibre sources, suggesting that a combination of insoluble fibre with a prebiotic fermentable fraction, having medium hydration properties, may improve performance in weaned piglets.

**Abstract:**

The effect of including micronised fibre sources (FS) differing in fermentability and hydration capacity (HC) on growth performance, faecal digestibility, and caecal fermentation was investigated in piglets. There were four dietary treatments: a control diet (CON) and three treatments differing in the HC and fermentability of FS added at 1.5% to prestarter (28–42 d) and starter (42–61 d) diets. These were: LHC (low-HC by-product-based insoluble fibre (IF) with a prebiotic fraction (PF) from chicory root); MHC (medium-HC by-product-based IF with a PF); and HHC (high-HC non-fermentable wood-based IF with no PF). There were eight replicates per treatment. Over the entire period, LHC and MHC piglets showed a 10% increase in daily growth and feed intake (*p* ≤ 0.019) and tended to have a reduced feed conversion ratio (*p* = 0.087) compared to HHC piglets. At 42 d, faecal protein digestibility increased by 5% in the LHC and MHC groups compared with the HHC group (*p* = 0.035) and did not differ from the CON group. Both LHC and MHC fibres were more fermented in vitro with caecal inocula from 61 d old piglets than HHC fibre (*p* ≤ 0.003). These results suggest that balanced soluble and insoluble fibre concentrates can improve piglet performance.

## 1. Introduction

In recent decades, dietary fibre (DF) has shifted from being considered an anti-nutritional factor to being recognised as an important dietary component that contributes to proper digestive physiology in non-ruminants [1,2]. Moderate levels of DF can enhance digestive health and prevent pathogen infection by directly influencing gut development, modulating the microbiome, and improving gut barrier integrity [3,4,5]. However, the mechanism of action and physiological effects of DF are highly dependent on its chemical composition (degree of lignification and types of polysaccharides), and physicochemical properties, such as solubility, viscosity, particle size, or hydration properties, which determine the fermentability [6,7,8,9]. DF is typically divided into two fractions based on solubility: insoluble and soluble fibre.

Sources of insoluble fibre (IF) may improve intestinal function, primarily due to their physical effect and several adaptations throughout the gastrointestinal tract [10]. They reduce the transit time [11] and modulate gut morphology [12]. For instance, they may prevent pathogenic species from adhering to the epithelial surface of the intestinal mucosa [13]. Among IF sources, lignocellulose is characterised by very low fermentability and high hydration capacity [14,15]. The inclusion of lignocellulose in piglet diets is common due to its positive effects on gut health and performance [16,17]. Although it has low fermentability, its 1% supplementation increased the ileal *Lactobacillus* population [16], and added at 1.5%, it increased caecal butyrate levels, providing energy for intestinal cells [1,18]. Supplementation at 2% increased the villus height to crypt depth ratio in the duodenum and jejunum, which is an indicator of intestinal mucosa maturity and functionality, relevant for nutrient absorption and barrier integrity [19]. Purified lignocellulose from sugarcane bagasse (1–3%) increased the average daily feed intake and improved the feed conversion ratio from 30 to 42 d [20]. Conversely, supplementation with 1% lignocellulose improved only the feed conversion ratio, whereas negative effects were observed on feed intake and body weight when supplemented at 0.8% [21].

By contrast, soluble fibre (SF) provides a more readily utilisable substrate for the intestinal microbiota, potentially improving gut health through the production of short-chain fatty acids [5,22], among other effects [23]. For instance, butyrate is the most important fermentation end-product involved in enhancing digestive tract functionality, maintaining gut integrity, and promoting host immunity, which may positively impact animal performance [24,25]. However, in weaned piglets, SF sources such as pectin or citrus pulp resulted in negative effects on gut morphology and poorer growth performance due to increased intestinal viscosity [7,12,26].

On the other hand, fructooligosaccharides (FOS) or inulin, typically obtained from chicory root, are sources of non-digestible oligosaccharides that are highly fermentable with low viscosity [27,28]. In vitro fermentation of chicory root showed an increase in the abundance of *Eubacterium* spp. and *Ruminococcus* spp., which was positively correlated with butyrate production [29]. Also, 0.4% addition of inulin with different polymerisation degrees (short-chain, long-chain, and both 50–50%) increased the abundance of *Bifidobacterium* spp. and *Lactobacillus* spp. and decreased less desirable populations of *Clostridia* and *Enterobacteriaceae* [30]. Supplementation with either 8% inulin or 0.6% FOS to piglets challenged with enterotoxigenic *E. coli* or sensitised with soybeans, respectively, reduced diarrhoea incidence [31,32], presumably due to FOS Nrf2 signalling activation and improvement in the expression of specific tight junction proteins [33]. In some cases, low inclusion levels also improved mucosa morphology: 0.25 and 0.50% inulin supplementation improved the villus height to crypt depth (V:C) in the duodenum and ileum and improved the distribution and abundance of the tight junction protein zonula occludens-1 in the duodenum and ileum epithelium [34]. All of these changes may contribute to maintaining barrier integrity and improve gut health. However, the positive effects of dietary inulin inclusion (0.25, 0.4, 0.5, and 1.0%) does not always produce effects on performance [34,35].

A combination of IF with prebiotic compounds can be a potential strategy to enhance intestinal fermentation, the gut barrier, and immunity, having potential synergic effects in the gut [36]. Using different proportions of lignocellulose and inulin in post-weaned piglets induced synergetic effects on blood biochemical indexes, nutrient digestibility, the hindgut microbiome, and gut barrier function [16], suggesting that a combination of IF and prebiotics provides an effective DF profile in the post-weaning period.

Industrial agrifood by-products are a suitable source of DF with bioactive properties [37,38]. However, some of them are difficult to use due to their hardness and size, but when properly milled, they can be included in the diet. Micronisation is a novel technology allowing particle size reduction to micrometre dimensions, which modifies the physical and chemical characteristics of fibre sources [39,40]. This process may improve the bioaccessibility and antioxidant capacity of IF [40,41] and increase the specific surface area, contributing to more extensive contact with the mucosal surface [42] and promoting higher crude protein digestibility [8]. Additionally, using these by-products in piglet feeding enhances farm sustainability and creates added value for these ingredients, increasing the food chain’s circularity, in line with current societal demands and polices. To the best of our knowledge, no previous work has evaluated the potential effects of mixtures of micronised IF from agricultural by-products and highly fermentable prebiotic sources in piglets. The objective of the current trial was to investigate the effects of low levels (1.5%) of inclusion of three micronised IFs, on performance, nutrient digestibility, and caecal fermentation in piglets during the post-weaning period. The micronised IFs used were the usual lignocellulose (high hydration capacity and low fermentability) and two new fibrous mixtures that were more fermentable and with medium or low hydration capacity.

## 2. Materials and Methods

### 2.1. Animal Ethics Statement

The experimental protocols were approved by the Ethics Committee of Animal Experimentation of Castilla y León Agriculture Technology Institute (2022/53/CEEA). Animal care and handling procedures were performed according to RD53/2013 [43].

### 2.2. Animals and Housing

The trial was conducted at the Porcine Testing Center of Castilla y León Agriculture Technology Institute in Hontalbilla (Segovia, Spain). A total of 192 piglets (Landrace × Duroc), 50% males and 50% females, weaned at 28 ± 1 d, came from a commercial farm (La Parilla, Valladolid, Spain) and were individually weighed (6.9 ± 1.12 kg) and marked before starting the experiment. 

There was a total of 32 pens (6 piglets/pen) distributed in 4 rooms with natural and artificial illumination (40 lux during 8 h daily). Each pen (2.46 × 1.54 m) had a plastic slotted floor and was equipped with a nipple bowl drinker and a metal feeder. The ambient temperature was maintained at 28 °C during the first week and was then gradually decreased by 2 °C every week.

### 2.3. Experimental Design and Diets 

Four experimental diets were prepared: a control diet (CON) with no additional fibre supplementation, and three diets based on the inclusion of 1.5% fibre sources differing in HC and fermentability. Fibre sources LHC and MHC were designed as a mixture of finely ground (micronised to a target size below 100 µm) IF from agricultural by-products. The low-HC (LHC) diet was formulated using only low hydration properties sources, such as almond shell, olive kernel, and nutshell (75%), and non-fermented grape pomace (15%). The medium-HC (MHC) diet incorporated 50% wood (*Pinus* spp., decorticated), 25% of a mixture of low-hydration property fibre sources (almond shell, olive kernel, and nutshell), and 15% non-fermented grape pomace. Both mixtures included chicory root (10%) to provide a fermentable fibre fraction. The high-HC (HHC) diet contained only wood IF, characterised as high-HC non-fermentable fibre. Fibre mixtures were added to both prestarter and starter diets. All of the sources used in the trial were finely ground, presenting geometric mean diameter (GMD) values of 13, 28, and 97 µm for LHC, MHC, and HHC, respectively. The chemical composition and physicochemical properties of the fibre sources included in the diets are summarised in Table 1.

The compositions and nutrient contents of the diets are shown in Table 2. All diets were formulated to be isocaloric and isonitrogenous, according to FEDNA (2013), to meet or exceed the nutrient recommendations for weaned piglets [44], and they were manufactured in pellet form (30 mm). The net energy (MJ/kg) content of the experimental diets was estimated based on the incorporation of each ingredient. The diets provided 11.1 MJ/kg in the prestarter diet and 11.4 MJ/kg in the starter diet, respectively.

The feed and fresh water were provided ad libitum. For preparation of experimental treatments, all raw materials were homogenised in a mixer (MMG 316, 250 L, Murcia, Spain) for 150 s. All diets contained 0.9% of diatomaceous earth (Celite ^®^) as an indigestible marker to determine the total tract apparent digestibility of nutrients [45,46]. Prestarter and starter diets were administrated from 28 to 42 d and 42 to 61 d of age, respectively.

### 2.4. Growth Performance Trial

A total of 96 males and 96 females were randomly assigned to one of the four dietary treatments according to homogeneity in the initial body weight (BW) among 32 pens (6 piglets/pen) distributed in 4 rooms. The distribution of the pigs was performed using the individual weight of the animals at weaning, grouped to keep similar animals in each box. All experimental animals were distributed between four rooms with 8 pens (4 for males and 4 for females): two rooms with small pigs (on average 5.7 kg) and two with large pigs (on average 7.8 kg). Within each room, the experimental treatments were randomly assigned between males and females to reach a similar initial body weight among treatments. Each treatment was replicated eight times, and the experimental unit was a pen with six piglets per pen. Individual BW of piglets and pen feed consumption were measured at 28 (weaning age), 42, and 60 d of age. The feed was provided from 25 kg bags. The feeders were filled twice a day to ensure enough feed throughout the day. The number of bags used and their weights were recorded daily. The consumption was calculated as a sum of the feed that disappeared during the prestarter (28–42), starter (42–61), and overall (28–61 d) period, divided between the number of piglets in the pen, and the days in each period. Data were used to calculate ADFI, ADG, and FCR. The morbidity and mortality were recorded daily.

### 2.5. Measurement of Total Tract Apparent Digestibility 

The total tract apparent digestibility (TTAD) of dry matter (DM), organic matter (OM), and crude protein (CP) was measured at 42 d and 61 d. The faecal samples were obtained by spontaneous defecation from three piglets per pen or the piglet chosen for slaughter at the end of prestarter and starter periods, respectively; faeces were then gently squeezed into plastic containers and frozen at −80 °C before being freeze-dried. Samples were ground using a centrifugal mill (Retsch Model Z-I, Stuttgart, Germany), and the concentration of acid-insoluble ash was measured in both the feed and faeces using the sequential method of Coca-Sinova et al. [45]. Briefly, DM was analysed by drying at 103 °C for 24 h, and then the samples were ashed by incineration at 600 °C for 12 h to determine the OM content. Then, 40 mL of 2 N HCl was added to each Erlenmeyer flask and the mixture was gently boiled for 5 min. The TTAD was calculated using the following equation:TTAD [%] = [1 − (Celite diet × Nutritional constituent faeces)/(Celite faeces × Nutritional constituent diet)] × 100
where ‘Celite diet’ and ‘Celite faeces’ represent the insoluble marker content in the diets and faeces, respectively. ‘Nutritional constituent faeces’ and ‘nutritional constituent diet’ represent the DM, OM, and CP content in the faeces and diets, respectively.

### 2.6. Gut Sampling and In Vitro Caecal Fermentation

At 61 d of age, one piglet per pen was pre-stunned with a captive-bolt gun and slaughtered by exsanguination via the jugular vein for intestinal digesta and tissue sampling. The pH of the digestive content of the ileum, caecum, and colon was measured immediately using a digital pH meter (model 507, Crison Instruments S.A., Barcelona, Spain). The whole caecum was weighed, placed in individual bags, and maintained at 4 °C until the in vitro fermentation trial was performed within the next 22 h. Samples of the ileum, colon, and faeces were dried at 70 °C for 72 h to determine the DM content. 

The caecal contents of two piglets from the same dietary treatment were pooled, and 5 g was weighed, mixed with 5 mL HCl 0.5 N, and frozen at −20 °C until the determination of the concentration of short-chain fatty acids (SCFAs). The rest of the caecal content was used as an inoculum for the in vitro trial to analyse the fermentative capacity of the piglet microbiota at 61 d.

### 2.7. In Vitro Caecal Fermentation

The three fibre sources used in the LHC, MHC, and HHC diets and their individual constituents (almond shell, olive kernel, wood, nutshell, grape pomace, and chicory root) were used as substrates for the in vitro incubations using the inoculum from piglets fed the CON diet. Additionally, LHC, MHC, and HHC fibre sources were used as the substrates to determine the impact of the adaptation of the caecal microbiota to fermentable fibre. The inoculum from piglets fed the CON diet was considered as ‘non-adapted microflora’ and that from animals fed each fibre source was considered as ‘fibre-adapted microflora’. 

The protocol was performed according to Ocasio-Vega et al. [47]. Briefly, 200 mg samples of DM of each substrate was weighed and placed into 60 mL glass vials. Three different inocula were used for each dietary treatment, each constituted by the pooled caecal content of two piglets, making a total of 12 inocula. Although initially four inocula per experimental treatment were collected, due to an accident in the laboratory, four inocula (one per treatment) were lost, and only three inocula per treatment were used. Two grams of each inoculum were mixed with 100 mL of the culture medium of Goering and Van Soest [48]. The mixture was homogenised with a blender (BP4570, 750 W, Ufesa, Vitoria-Gasteiz, Spain) for 20 s and filtered through a double piece of clean cheesecloth. Vials were filled with 50 mL of the mixture using a peristaltic pump (Watson-Marlow 520UIP31; Watson-Marlow Fluid Technology Group, Falmouth, UK) under continuous flushing with CO_2_, sealed with rubber stoppers, and incubated at 40 °C for 72 h. All substrates were incubated with inocula from all dietary treatments. 

A total of 120 vials, 108 with substrate and 12 without substrate (blanks; one per inoculum), were incubated. Blanks were incubated to correct for gas production by the inoculum. Total gas production was quantified by measuring the gas produced using a digital pressure gauge (HD 2304.0, Delta OHM, Caselle di Selvazzano, Italy) and a plastic syringe at 3, 9, 24, 33, 48, and 72 h of incubation. The gas produced was released after each measurement.

### 2.8. Laboratory Analysis

#### 2.8.1. Chemical Composition

Diets and faeces were ground through a 1 mm screen using a centrifugal mill (Retsch Model Z-I, Stuttgart, Germany) and then analysed using AOAC [49] procedures to determine dry matter (DM, method 934.01), ash (method 942.05), ether extract (EE, 920.39), and nitrogen by Dumas (method 968.06) using a Leco analyser (model FP-528; Leco Corp., St. Joseph, MI, USA). The CP content was determined by multiplying the nitrogen content by 6.25. Insoluble and soluble fibre were determined according to the AOAC 991.43 protocol using Fibertec^®^ 1023 (FOSS System, Hilleroed, Denmark), and both values were added to obtain the total dietary fibre (TDF) content. Dietary neutral detergent fibre (NDF), acid detergent fibre (ADF), and acid detergent lignin (ADL) were determined sequentially using the filter bag system with F57 bags with a pore size of 25 µm (Ankom Technology, New York, NY, USA) by adapting the methods of Horwitz [50] and Mertens [51]. Thermo-stable alpha-amylase (Ankom Technology, New York, NY, USA) and sodium sulphite (PanReac Appli Chem, 131,717.1211,Darmstadt, Germany) were used, and the value was corrected for ash content. 

#### 2.8.2. Physicochemical Properties

The particle size, expressed as GMD, and the particle size distribution of different fibre sources were determined in 100 g samples using a sieve shaker (FTS-0200, Filtra, Badalona, Spain) equipped with 5 sieves ranging in mesh from 62 to 1000 μm (>1000, 500–1000, 250–500; 250–105; 105–62; <62 μm; ASAE, 1995 [52]). The hydration capacity of the fibre sources was measured by determining the water-binding capacity (WBC, g/g) and the swelling capacity (SC, g/mL) by adapting the methods described by Slama et al., Berrocoso et al., and Priester et al. [14,53,54]. Briefly, 0.4 g of each fibre source was hydrated for 22 h with 10 g of water, then centrifuged at 3100× *g* for 20 min (5810R Centrifuge, Eppendorf, Wesseling-Berzdorf, Germany). The unabsorbed water was weighed, and the WBC was calculated as the difference between 10 g of water and the supernatant (g) divided by the starting weight of the sample. The SC was measured using 1 g of sample, which was gently stirred, incubated with 20 mL of water, then left in a graduated metric-scale cylinder for 22 h. The final volume (mL) of the sample was divided by the starting weight (g). All analyses were performed in triplicate.

#### 2.8.3. Short-Chain Fatty Acid Determination 

Caecal content processing for determining SCFA concentrations was adapted from Kimiaeitalab et al. [55]. A 5 g sample of the caecal mixture was mixed with 5 mL of 0.5 M HCl, homogenised, and centrifuged at 13,000× *g* for 15 min at 4 °C. Then, 1 mL of the supernatant was mixed with 0.5 mL of a deproteinising solution (20 g metaphosphoric acid and 0.6 g of crotonic acid/L) and left overnight at 4 °C. The total SCFA concentration was determined by gas chromatography using a Shimadzu GC 2010 chromatograph (Shimazdu Europa GmbH, Duisburg, Germany) equipped with a TR-FFAP column (30 m × 0.53 mm × 1 µm; Supelco, Madrid, Spain). Individual SCFAs were identified according to the procedure described by García-Martínez et al. [56].

### 2.9. Statistical Analysis

Statistical analyses were performed using SPSS (IBM SPSS Statistics for Windows, Version 26.0. IBM Corp., Armonk, NY, USA). The effects of the dietary treatments were analysed by one-way analysis of variance (ANOVA), and the model included the diet (D) as the main effect. The model for growth traits also included the average BW at weaning (BW0) of the pen as a covariate, as follows:Y_ij_ = µ + D_i_ + β (BW0_ij_ − average of BW0) + ε_ij_

The residues of the model for each trait were analysed to confirm that they were normally distributed (Shapiro–Wilk or Kolmogorov–Smirnov tests), and the homoscedasticity was evaluated from the graph of residue distribution and confirmed with the Levene’s test. Furthermore, orthogonal contrasts were applied to study: C1: differences between CON and fibre-supplemented animals (CON vs. LHC + MHC + HHC); C2: differences between high and low fermentable fibre sources (LHC + MHC vs. HHC); and C3: differences between both fibre sources containing a fermentable fraction (LHC vs. MHC). Linear and quadratic regressions between the WBC of the fibre sources and growth performance were determined. The morbidity and mortality rates were analysed by Pearson′s chi-square test. 

In vitro gas production data were analysed individually for each measurement time and independently for each preplanned comparison. Differences in gas production among dietary fibre sources (LHC, MHC, and HHC) were tested using the caecal content from CON-fed piglets as the inoculum, which was considered to be an inoculum non-adapted to fibre. The potential effects of the experimental diets on gas production were tested independently for each fibre source used as a substrate in the in vitro trial.

Piglet performance data were expressed as least-squares means, whereas all other variables were reported as means with standard error of the mean (SEM). For all statistical analyses, effects were considered significant when *p* < 0.050, whereas a trend was declared when 0.050 ≤ *p* ≤ 0.100. When a significant effect was detected, means multiple comparisons were carried out using the Tukey test.

## 3. Results

### 3.1. Growth Traits

From 28 to 42 d of age, piglets fed LHC and MHC diets grew on average 10% faster than those in the HHC group and consumed 7% more feed (*p* ≤ 0.036; Table 3). During this period, the MHC group tended to have a higher ADG than the LHC group (*p* = 0.052), but their ADG and ADFI were similar to the CON group. Dietary treatments had no effect on FCR during this period (*p* ≥ 0.15). From 42 to 60 d of age, both LHC and MHC groups grew on average 12% faster and consumed 10% more feed (*p* ≤ 0.025) than piglets fed the HHC diet. Moreover, the MHC group tended to have a higher ADFI (*p* = 0.064) and similar ADG to LHC-fed piglets but had higher ADG and ADFI than the CON group (by 7 and 12%, respectively; *p* < 0.05). By contrast, diet had no effect on FCR (*p* ≥ 0.11) during this period. When the entire experimental period (28–60 d of age) was considered, piglets fed LHC and MHC diets had 10% increases in both ADG and ADFI compared with those fed the HHC diet (*p* ≤ 0.019) and tended to have a reduced FCR (*p* = 0.087). Piglets fed the MHC diet had 9% higher ADG and ADFI than the CON group and grew 6% faster than the LHC group (*p* < 0.05). 

In the current trial, piglets did not exhibit severe digestive disorder. However, during the starter period, some cases of meningitis and lameness were reported (affecting 23.4, 8.3, 12.5, and 27.7% of piglets in the CON, LHC, MHC, and HHC groups, respectively, *p* = 0.047), and piglets were treated individually with amoxycillin and glucocorticoids (Bivamox ^®^ 150 mg/mL, Caliercortin ^®^ 4 mg/mL). All of the animals remained in the experiment in accordance with the veterinarian’s decision, based on their positive response to medical treatment and recovery from clinical symptoms.

### 3.2. Total Tract Apparent Digestibility 

At 42 d, the total tract apparent CP digestibility of LHC and MHC piglets was 5% higher than that in the HHC group (*p* = 0.035), and a similar trend was observed for DM and OM digestibility (*p* ≤ 0.098), but values did not differ from those in the CON group (Table 4). Dietary treatment had no effect on the total tract apparent digestibility of DM, OM, and CP at 61 d of age (*p* ≥ 0.20).

### 3.3. Gut Traits and Short-Chain Fatty Acid Concentration

The pH of the ileal digesta in the fibre-supplemented groups (LHC, MHC, and HHC) was 5% higher compared to that in the CON group (*p* = 0.045; Table 5). Piglets fed the MHC diet tended to have a lower ileal pH compared to the LHC group (*p* = 0.087). Dietary treatments had a low impact on caecal SCFA concentrations at 61 d of age. Piglets fed the MHC diet tended to have lower total SCFA concentrations than those fed the LHC diet (*p* = 0.080), which was due to a reduction in propionate (*p* = 0.044) and butyrate (*p* = 0.076) levels. No other effects of the dietary treatments were observed on caecal SCFA concentrations, either when expressed as µmol/g or as molar proportions (*p* ≥ 0.12). Dietary treatment had no effect on the pH of the caecum and colon (*p* ≥ 0.15), caecal weight (*p* ≥ 0.36), or moisture content of the digesta (*p* ≥ 0.17. Figure 1).

The cumulative gas production curves of different micronised raw materials and the experimental fibre sources (LHC, MHC, and HHC) when the caecal content of CON-fed piglets was used as the inoculum are presented in Figure 2A,B, respectively. As expected, chicory root showed the highest gas production, whereas grape pomace, olive kernel, almond shell, nutshell, and wood generated lower amounts of gas at all incubation times (*p* < 0.001). Differences among these IF sources were detected only at 9 h, when grape pomace produced more gas than the rest of the substrates (Figure 2A). Fibre sources included in both the LHC and MHC diets showed higher gas production than the HHC diet (*p* ≤ 0.003; Figure 2B) from 24 h of incubation onward, but no differences were detected at shorter incubation times (3 and 9 h). Generally, gas production reached the maximal value at 33 h, with the only exception being chicory root, which continued generating gas until the end of the incubation at 72 h.

The gas production curves of the fibre sources included in the LHC, MHC, and HHC diets when caecal inocula from piglets fed the CON, LHC, MHC, and HHC diets were used are presented in Figure 3A–C, respectively. A tendency for greater gas production from 33 h onward in the non-adapted microbiota in the LHC (*p* = 0.082) and MHC (*p* = 0.061) groups was observed. No effect of adaptation was observed in the HHC group (*p* ≥ 0.25).

## 4. Discussion

The TDF values of the analysed samples may have been underestimated for LHC and MHC, as the SF content came mostly from inulin, which cannot be precipitated with ethanol using the SF method [57]. Instead, NDFom analysis was performed using F57 Ankom filter bags that retained only particles >25 μm. Therefore, loss of smaller particles from the bags was possible [58], and these losses may vary depending on the fibre source. Analysis of the water binding and swelling properties confirmed that the LHC and MHC sources were properly designed with low and medium HC, respectively.

### 4.1. Growth Performance and Diet Digestibility

Animals fed HHC performed similarly to those fed the control diet, but both the control and HHC groups had lower performance than the MHC group. This may be associated with the lower feed intake registered in these groups during the experimental period and lower CP digestibility at 42 d, although FCR was not significantly affected. In general, the results of this trial seem to confirm that a combination of the physical effects of insoluble fibre with highly fermentable sources may be advantageous for performance. The results partially align with what was previously reported by Chen et al. [16], who observed an improvement in the FCR of animals supplemented with 0.5% inulin and 0.5% lignocellulose at 38 and 52 d compared to control animals. However, Molist et al. [59] also supplemented piglet diets with insoluble and soluble fibre sources, based on wheat bran and sugar beet pulp, but no effect was observed on performance. This may indicate that the inclusion of fermentable fibrous sources that do not increase intestinal viscosity may be more recommended for young piglets.

On the other hand, micronised wood-lignocellulose has been widely used in recent investigations as an IF source with generally positive effects, although some contradictory results have also been reported. Supporting our results, supplementation with 1% non-fermentable lignocellulose [16] or 1.5% non-fermentable or partially fermentable lignocellulose [18] produced no effects on the ADG or ADFI of weaned piglets. Other authors reported that 1% supplementation of lignocellulose increased both ADFI and ADG in piglets from 31 to 52 d, leading to a higher final body weight compared to the control diet [60]. These inconsistent results suggest that dosage, the interaction with other dietary constituents, the age, and the timing of ingestion of highly lignified fibre sources may affect the performance of weaned piglets.

Better ADG and ADFI in MHC animals compared to those fed IF lignocellulose aligned with previous results observed in piglets fed both inulin and lignocellulose, suggesting synergistic effects of a combination of IF and prebiotic sources on piglet performance in the post-weaning period [16]. The inclusion of 0.5% inulin has been reported to improve the growth rate, which was associated with an increase in serum levels of IGF-1, regulating growth hormone and stimulating cell proliferation in both weaning and growing pigs [34,61]. 

In the current trial, MHC increased CP digestibility at 42 d compared to HHC, which likely had a direct impact on growth performance at this stage. All fibre sources used in our study were highly lignified, suggesting low fermentability. However, the inclusion of a fermentable fraction in LHC and MHC diets was designed to provide a usable substrate to the resident microbiota. It was hypothesised that the combined supplementation of IF with a soluble fraction may improve gut health by beneficial modifications of microbial colonisation and fermentation patterns [59], leading to more efficient nutrient utilisation. 

The positive correlation observed between fermentability and nutrient digestibility may be attributed to a longer retention time of the digesta and greater enzymatic secretion, both leading to increased subsequent nutrient absorption [62]. In weaned piglets, 0.5% inulin supplementation increased the expression levels of GLUT2 and DMT1, associated with higher digestive capacity in the proximal intestinal mucosa [34].

In addition, the particle size of all experimental treatments in our study was particularly low and provided a high number of inert particles in contact with the intestinal mucosa, potentially benefiting gut physiology or morphology. In fact, previous studies reported that wheat bran supplemented at 3% increased both the villus height to crypt depth ratio in the ileum and sucrase and maltase activities in the intestine of piglets [63]. Similarly, increasing levels of inulin supplementation (0.5–2 g/d) during the suckling period increased the villus height and the villus height to crypt depth ratio in the jejunum and ileum of piglets after 28 d [64]. Although previous studies reported none or even negative effects on CP digestibility, depending on the inclusion level of fibre sources [65,66], a higher absorption surface due to possible changes in mucosa morphology may explain the greater digestibility observed in our study for the MHC diet.

Dietary treatment showed no effect on caecal weight or the moisture content in different intestinal compartments. The moisture content of the digesta observed in this research generally aligned with previous reports [67]. The moisture of digestive contents is a balance between the secretion and absorption of water from the duodenum to the distal colon, crucial for maintaining epithelium hydration and correct functionality [5].

Higher CP digestibility observed for the MHC diet at 42 d may be associated with potential improvements in digestive efficiency, leading to better performance in the postweaning period compared to lignocellulose inclusion, although no changes were observed in the digestive traits or moisture levels of the digestive contents. In the current trial, the hydration capacity of the treatments had no linear or quadratic effect on performance (*p* ≥ 0.237, r = 0.215, *n =* 24).

### 4.2. In Vitro Fermentation and Caecal Content Characteristics

Dietary fibre plays an crucial role in maintaining gut microbiota balance and gut health [5,22]. Various pathogenic bacteria species, such as *Escherichia coli, Klebsiella* spp., *Campylobacter* spp., *Streptococcus* spp., *Clostridium perfringens*, *Clostridium difficile*, and *Bacteroides fragilis*, produce harmful metabolites that can impair barrier function, leading to colonic diseases and diarrhoea problems [2,68]. Conversely, a higher abundance of *Roseburia*, *Prevotella*, and genera belonging to *Ruminococcaceae*, which are adapted to metabolise complex oligosaccharides and polysaccharides, may protect the host from pathogen infections [69]. 

Fibre supplementation in the diet alters the richness of the microbiota, although the proportions and diversity of fibre-degrading bacteria depend on the structure and composition of the fibre sources [70,71]. For instance, feeding 1% inulin resulted in greater microbial diversity and richness compared to the same level of lignocellulose [72]. A combination of 4% wheat bran and 3% sugar beet pulp reduced enterobacteria counts in the digesta, indicating a synergistic effect of the two different sources on the microbial population [59]. Additionally, dietary supplementation with 0.5% lignocellulose and 0.5% inulin increased the relative abundance of the phylum Bacteroidetes and the genera *Selenomonas*, *Phascolarctobacterium*, *Sharpea*, and *Alloprevotella* [72]. 

In this study, the microbiota was not analysed directly. However, all of the fibre sources were highly lignified, and fine grinding was performed to increase the surface area for microbial colonisation [73]. Furthermore, the inclusion of inulin from chicory root would likely have a positive impact on the fibre-degradation microbiota in LHC and MHC piglets.

The individual fibre sources—almond shell, olive kernel, wood, and nutshell—showed cumulative gas production below 10 mL/g DM after 72 h of incubation with CON inoculum. Only grape pomace had slightly greater gas production at short incubation times, reaching 14 mL/g DM, likely due to its moderate protein content [74], although no increase in gas production was observed after 24 h of incubation. By contrast, chicory root, which has a high content of inulin [75], demonstrated much greater and prolonged gas production, suggesting its potential use as a prebiotic source [76]. 

The HHC fibre source showed low gas production (<3 mL/g DM), consistent with previously reported results using faeces or caecal digesta of pigs as the inoculum [15,77]. The greater gas production of LHC and MHC fibre sources (>14 mL/g DM) confirmed the potential fermentation capacity of their fermentable fraction. These results confirm the low fermentability of lignocellulosic fibre sources. All of these results were in line with our previous study in 21 d [78] and 42 d [46] broilers, confirming the low fermentability of lignocellulosic fibre sources. 

Unexpectedly, the fibre sources used in the current experiment had no impact on caecal SCFA concentrations, despite differences observed in their fermentation in the in vitro trial. A lack of effects on caecal characteristics was also previously reported with the supplementation of cellulose at 0.5% or 2% [79] or finely ground wheat bran at 4% [6] to piglets. However, lignocellulose addition at 1, 1.5, 2, and 3% improved caecal butyrate formation in another studies [18,20]. Also, a combination of insoluble and soluble fibre based on wheat bran and sugar beet pulp, or lignocellulose with inulin, promoted a beneficial shift in microbial colonisation, with a higher production of butyric acid in the large intestine [16,59]. Therefore, the lack of effects may be related to diet composition, fibre inclusion level, transit time, fibre composition, or interactions among all of them.

In our study, it was expected that the prebiotic fraction from chicory root would decrease the caecal pH and increase SCFA production. Chicory root contains inulin and fructans and was included in the LHC and MHC diets to create the properties of an ideal fibre source, given its potential to modify the intestinal microbial profile [28]. Surprisingly, the results showed no impact of inulin addition on the fermentation patterns in the caecum. 

The effect of inulin supplementation on SCFAs are ambiguous. Supplementation with 0.25% inulin did not affect SCFA levels in the caecum and colon, although ileal propionate levels were increased compared to the same level of cellulose inclusion [80]. High dietary inulin (3%) resulted in lower acetate but higher proportion of propionate and butyrate levels compared to pigs fed wheat bran [81]. According to a metanalysis [66], digesta pH in the ileum, caecum, colon, and faeces, as well as SCFA concentrations in the gastrointestinal tract and faeces, appeared to be largely unaffected by dietary inulin, which is consistent with our results. 

The lack of difference in in vivo caecal fermentation may be attributed to factors such as the source of inulin, the level of supplementation, or the degree of polymerisation, which suggest extensive pre-caecal fermentation [82,83]. Short-chain inulin may be degraded in the jejunum, whereas long-chain inulin has been detected in all intestinal segments [81,84]. Studies involving post-weaned piglets fed different types of inulin at 4% of the diet reported an absence of inulin in digesta samples from the caecum as well as the proximal, mid, and distal colon [85]. The loss of fructans during passage through the small intestine could be due to hydrolysis by acid or enzymatic degradation by the microflora present in the small intestine [86]. The discrepancies observed among studies may also be related to factors such as the inclusion level of fibre sources, the composition of the basal diet, and the characteristics of the target animals, including age and the maturity stage of intestinal tract development [87,88].

Since the fibre sources used in our study might influence the microbiota profile due to their adaptation to fermenting different types of fibre [89], it was hypothesised that exposure to each fibre source during the post-weaning period would increase the fermentative capacity of the microbiota. Unexpectedly, both LHC and MHC tended to produce more gas when incubated with an inoculum from piglets fed the control diet than with an inoculum from piglets fed the corresponding diet (‘adapted inoculum’). This finding suggests that, as indicated by the in vivo results, inulin was likely fermented in the proximal intestine and did not reach the caecum. Consequently, the caecal inoculum was probably not truly ‘adapted’, as the fibre had already been fermented in a previous part of the intestine. 

On the other hand, gas production from the HHC substrate was unaffected by microbiota exposure. The results suggest that inulin is easily fermentable, and prior adaptation of the microbiota is not necessary for efficient degradation. However, this appears to differ for other types of substrates. When wheat starch was incubated with faecal inoculum from weaned piglets, it showed greater fermentation (faster gas production and more SCFA production) using an inoculum from animals fed a control diet than using an inoculum previously ‘adapted’, although sugar beet pulp was better fermented using the adapted inoculum [90]. 

In summary, the high fermentability of LHC and MHC observed in the in vitro fer-mentation trial confirms a synergetic effect between insoluble fibre with a prebiotic fraction, which positively influenced performance and crude protein digestibility. Fibre fermentation is generally a beneficial process, mediated by the proliferation of saccharolytic bacteria [68], which helps to prevent the growth of facultative pathogens [1]. The synergetic supplementation of insoluble and highly fermentable fibre may provide the optimal balance for the gut microbiota, which could be crucial for modulating intestinal permeability and maintaining gut barrier integrity. 

Due to several limitations of this study, further research is needed to explore: (i) the impact of different combinations of fibre sources, (ii) varying levels of hydration capacity or fermentability, (iii) different levels of fibre inclusion, and (iv) additional physiological parameters or markers that could enhance our understanding of fibre’s role in physiology, digestibility, fermentation, and gut health.

## 5. Conclusions

This investigation supports the idea that micronisation of highly lignified agricultural by-products, such as almond shell, olive kernel, nutshell, or grape pomace, may provide an alternative fibre source for animal nutrition. Achieving an optimal balance between soluble and insoluble fibre with medium hydration properties could be a practical strategy for maintaining post-weaning performance in piglets. In this case, the combination of insoluble fibre with a prebiotic fraction from chicory root provided advantages in performance and crude protein digestibility compared with lignocellulose at 1.5% inclusion in the diet. Further research is needed to elucidate the impact of the hydration capacity of insoluble fibre sources in piglets.

## Figures and Tables

**Figure 1 animals-14-02612-f001:**
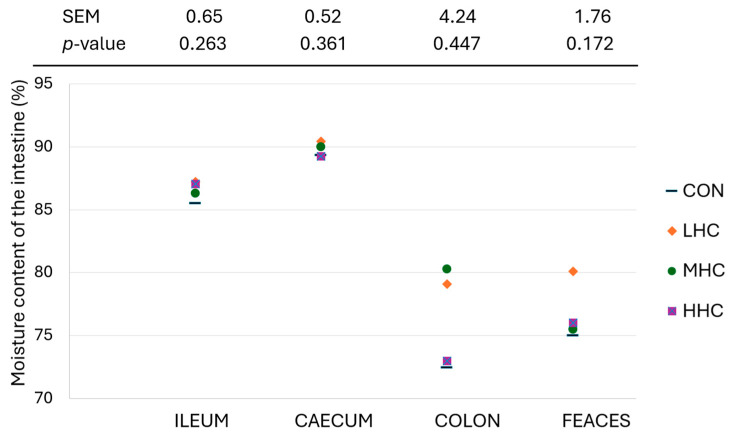
Effect of dietary treatment on the moisture content of the intestinal digesta (ileum, caecum, colon, and faeces) of 61 d old piglets. CON: basal diet with no additional fibre inclusion, LHC: basal diet including 1.5% low-hydration capacity insoluble fibre with fermentable fraction, MHC: basal diet including 1.5% medium-hydration capacity insoluble fibre with fermentable fraction, and HHC: basal diet including 1.5% high-hydration capacity insoluble fibre; *n* = 8 for all treatments.

**Figure 2 animals-14-02612-f002:**
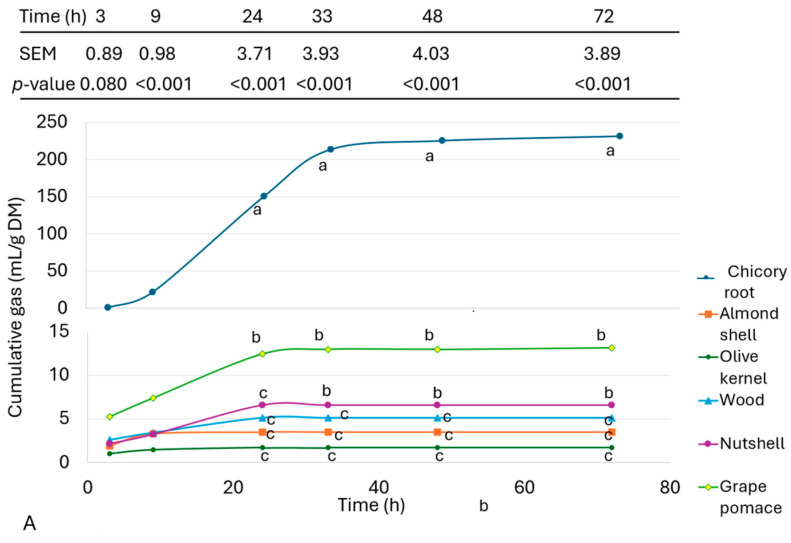
Cumulative gas production curves (mL/g DM) of (**A**) LHC, MHC, HHC, and (**B**) different micronised fibre sources (chicory root, almond shell, olive kernel, wood, nutshell, and grape by-products) after their incubation with the caecal content from 61 d old piglets fed the control diet. Three different inocula were used, and each inoculum was pooled from the caecal content of 2 piglets fed the control diet. a, b, c: Within the same row, means with different letters differ (*p* < 0.05; Tukey test). Values in the table indicate the SEM (*n* = 3) and *p*-value of the ANOVA analysing potential differences in gas production at each measurement time.

**Figure 3 animals-14-02612-f003:**
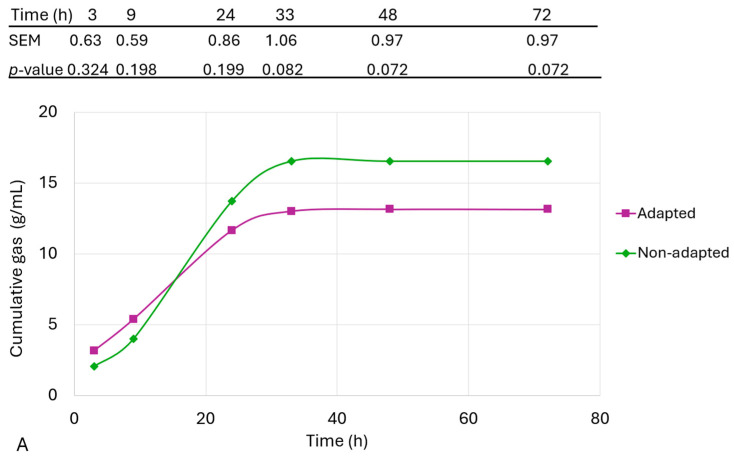
Cumulative gas production curves (mL/g DM) of (**A**) LHC: low-hydration capacity insoluble fibre with fermentable fraction, (**B**) MHC: medium-hydration capacity insoluble fibre with fermentable fraction, and (**C**) HHC: high-hydration capacity but non-fermentable insoluble fibre, when they were fermented either using caecal inoculum from 61 d old piglets fed a diet including the incubated fibre source (adapted microbiota) or a control diet without fibre sources (non-adapted microbiota). Three different inocula were used for each experimental dietary treatment, and each inoculum was the pooled caecal content from 2 piglets fed the same diet. Tables indicate the SEM (*n =* 3) and *p*-value of the ANOVA analysing potential differences in gas production values at each measurement time.

**Table 1 animals-14-02612-t001:** Fibre composition (%, as fed basis) and hydration properties characterised by water binding (WBC, g/g) and swelling (SC, g/mL) capacities of fibre sources included in the experimental diets ^1^.

	Fibre Sources Used in Experimental Diets		
	LHC	MHC	HHC
Chemical composition			
Dry matter	92.3	92.6	92.3
Total dietary fibre	79.5	82.7	90.5
Soluble fibre	2.5	2.7	1.0
Insoluble fibre	77.0	80.0	89.5
Neutral detergent fibre (NDF)	72.4	72.9	87.8
Acid detergent fibre (ADF)	49.9	55.5	75.8
Acid detergent lignin (ADL)	18.5	21.0	25.2
Hemicellulose ^2^	22.5	17.4	18.6
Cellulose ^3^	31.4	34.5	43.2
Crude protein	4.95	4.72	1.22
Hydration capacity			
WBC, g/g	2.55	3.97	6.54
SC, ml/g	3.99	5.51	7.17
Geometric mean diameter (GMD), µm	12.9	28.0	97.0

^1^ LHC: low-hydration capacity (HC) insoluble fibre with fermentable fraction, MHC: medium-HC insoluble fibre with fermentable fraction, and HHC: high-HC insoluble fibre as 100% micronised wood; ^2^ As NDF—ADF; ^3^ As ADF—ADL.

**Table 2 animals-14-02612-t002:** Ingredients (%, as fed basis) and calculated chemical compositions of experimental diets ^1^.

	Prestarter Diet				Starter Diet			
Ingredients	CON	LHC	MHC	HHC	CON	LHC	MHC	HHC
Barley	20.30	20.00	20.00	20.00	---	---	---	---
Wheat	19.80	17.60	17.60	17.60	8.70	5.80	5.80	5.80
Corn	---	---	---	---	44.94	44.90	44.90	44.90
Extruded corn	15.00	15.00	15.00	15.00	---	---	---	---
Cookie meal	16.50	16.50	16.50	16.50	20.00	20.00	20.00	20.00
Sweet whey	10.00	10.00	10.00	10.00	---	---	---	---
Soy protein concentrate	4.40	4.90	4.90	4.90	1.30	0.85	0.85	0.85
Fish meal, CP 64%	4.00	4.00	4.00	4.00	1.50	1.50	1.50	1.50
Soybean meal, CP 48%	2.50	2.50	2.50	2.50	17.00	18.25	18.25	18.25
Spray-dried porcine plasma	2.00	2.00	2.00	2.00	---	---	---	---
Low-HC fibre source	---	1.50	---	---	---	1.50	----	----
Medium-HC fibre source	---	---	1.50	---	---	---	1.50	----
High-HC fibre source	---	---	---	1.50	---	---	---	1.50
Celite ^®^	0.90	0.90	0.90	0.90	0.90	0.90	0.90	0.90
Calcium formate	0.79	0.79	0.79	0.79	---	---	---	---
Calcium carbonate	---	---	---	---	0.93	0.93	0.93	0.93
Monocalcium phosphate	0.70	0.70	0.70	0.70	1.12	1.12	1.12	1.12
Vitamin/mineral premix ^2^	0.50	0.50	0.50	0.50	0.50	0.50	0.50	0.50
MCFA monoglyceride mix ^3^	0.40	0.40	0.40	0.40	0.30	0.30	0.30	0.30
Soybean oil	0.32	0.85	0.85	0.85	0.80	1.45	1.45	1.45
Organic acid mix ^4^	0.30	0.30	0.30	0.30	0.30	0.30	0.30	0.30
Benzoic acid	0.25	0.25	0.25	0.25	---	---	---	---
Salt	0.25	0.25	0.25	0.25	0.43	0.44	0.44	0.44
L-Lysine HCl, 78.8%	0.48	0.47	0.47	0.47	0.57	0.55	0.55	0.55
L-Threonine, 98%	0.23	0.23	0.23	0.23	0.26	0.26	0.26	0.26
DL-Methionine, 99%	0.18	0.18	0.18	0.18	0.21	0.21	0.21	0.21
L-Valine, 98%	0.09	0.09	0.09	0.09	0.11	0.11	0.11	0.11
L-Tryptophan, 98%	0.06	0.06	0.06	0.06	0.08	0.08	0.08	0.08
Choline chloride	0.05	0.05	0.05	0.05	0.05	0.05	0.05	0.05
Nutritive value, % DM								
dLys ^5^	1.40	1.40	1.40	1.40	1.40	1.39	1.39	1.39
Estimated net energy (MJ/kg)	11.1	11.1	11.1	11.1	11.4	11.4	11.4	11.4
Analysed composition, % DM								
Organic matter	92.8	92.3	91.9	92.1	92.8	92.2	92.4	93.1
Crude protein	20.7	20.9	21.0	20.7	20.0	20.4	20.2	19.9
Neutral detergent fibre	12.2	13.6	13.9	13.8	12.0	13.5	13.5	13.9
Acid detergent fibre	3.7	4.2	4.1	4.4	3.5	3.9	4.0	4.2
Acid detergent lignin	0.4	0.6	0.6	0.6	0.3	0.5	0.5	0.9

^1^ CON: basal diet with no additional fibre inclusion, LHC: low-hydration capacity (HC) insoluble fibre with fermentable fraction, MHC medium-HC insoluble fibre with fermentable fraction, and HHC: high-HC non-fermentable insoluble fibre. ^2^ Provided per 1 kg of the diet: vitamin A (3a572a) 15,000 IU, vitamin D3 (3a671) 2000 IU, vitamin E (3a700) 250 mg, vitamin K3 (3a711) 2 mg, folic acid (3a316) 50 mg, niacinamide (3a315) 10 mg, calcium D-pantothenate (3a341) 10 mg, vitamin B1 (3a521) 10 mg, vitamin B2 (3a625) 10 mg, vitamin B6/pyridoxine hydrochloride (3a831) 10 mg, vitamin B12/cyanocobalamin 0.05 mg, biotin (3a880) 0.2 mg, iron sulphate (II) monohydrate (3b103) 375 mg, copper oxide (I) (3b412) 131 mg, manganese oxide (II) (3b502) 77.6 mg, zinc oxide (3b603) 150 mg, sodium selenite (3b801) 0.3 mg, anhydrous calcium iodate-I (3b202) 1.5 mg. ^3^ ENTERO-Nova MTB 400G+, Eastman-3F Feed and Food. ^4^ Formic and propionic mixture, Eastman-3F Feed and Food. ^5^ Standardised ileal digestible lysine. Remaining amino acids were adjusted to: Met 38%, Met + cys: 60%, Thr: 65%, Trp: 20%, Val: 69%.

**Table 3 animals-14-02612-t003:** Effect of dietary treatments on body weight (BW) and performance of piglets during the prestarter, starter, and overall post-weaning period.

	Dietary Treatment (DT) ^1^						*p*-Value ^3^			
Item	CON	LHC	MHC	HHC	SEM ^2^	COV	DT	C1	C2	C3
Prestarter (28–42 d)										
ADFI, g	356 ^a^	334 ^ab^	352 ^a^	319 ^b^	10.2	<0.001	0.033	0.119	0.036	0.112
ADG, g	316 ^a^	307 ^ab^	332 ^a^	291 ^b^	9.9	<0.001	0.027	0.670	0.018	0.052
FCR	1.125	1.092	1.064	1.098	0.02	0.48	0.401	0.149	0.551	0.502
BW 42 d	11.1 ^ab^	11.0 ^ab^	11.4 ^a^	10.8 ^b^	0.14	<0.001	0.028	0.915	0.017	0.053
Starter (42–60 d)										
ADFI, g	812 ^b^	836 ^ab^	912 ^a^	792 ^b^	21.3	<0.001	0.028	0.291	0.025	0.064
ADG, g	539 ^bc^	561 ^ab^	578 ^a^	510 ^c^	12.0	<0.001	0.002	0.431	<0.001	0.301
FCR	1.510	1.493	1.512	1.561	0.03	0.083	0.386	0.729	0.108	0.660
BW 60 d	20.8 ^bc^	21.1 ^b^	22.0 ^a^	20.00 ^c^	0.31	<0.001	0.001	0.561	<0.001	0.049
Overall (28–60 d)										
ADFI, g	632 ^b^	636 ^ab^	691 ^a^	603 ^b^	15.6	<0.001	0.029	0.610	0.019	0.056
ADG, g	455 ^bc^	464 ^b^	493 ^a^	429 ^c^	9.98	<0.001	0.001	0.561	<0.001	0.049
FCR	1.388	1.372	1.358	1.411	0.02	0.278	0.355	0.753	0.087	0.647

^a,b,c^: Within the same row, means with different letters differ (*p* < 0.05; Tukey test). ^1^ CON: basal diet with no additional fibre inclusion, LHC: basal diet including 1.5% low-hydration capacity insoluble fibre with fermentable fraction, MHC: basal diet including 1.5% medium-hydration capacity insoluble fibre with fermentable fraction, and HHC: basal diet including 1.5% high-hydration capacity insoluble fibre; ^2^
*n* = 8; ^3^ COV: Weaning body weight (6.7 ± 1.12 kg) was used as a covariate; C1: CON vs. LHC, MHC and HHC; C2: LHC and MHC vs. HHC; C3: LHC vs. MHC.

**Table 4 animals-14-02612-t004:** Effect of dietary treatments on total tract apparent digestibility (TTAD) of dry matter (DM), crude protein (CP), and organic matter (OM) in piglets at 42 and 61 d of age.

	Dietary Treatment (DT) ^1^					*p*-Value ^3^			
Item	CON	LHC	MHC	HHC	SEM ^2^	DT	C1	C2	C3
TTAD, 42 d									
DM	82.4	81.4	83.6	80.3	0.98	0.152	0.596	0.098	0.541
CP	80.9 ^ab^	80.0 ^ab^	83.9 ^a^	78.3 ^b^	1.26	0.040	0.936	0.035	0.802
OM	84.7	83.8	85.7	82.7	0.87	0.139	0.555	0.076	0.789
TTAD, 61 d									
DM	82.5	82.7	82.0	82.8	0.65	0.712	0.207	0.59	0.440
CP	77.4	76.7	78.3	78.9	1.17	0.566	0.697	0.34	0.337
OM	84.1	85.2	84.7	84.9	0.58	0.557	0.200	0.95	0.538

^a,b^: Within the same row, means with different letters differ (*p* < 0.05; Tukey test). ^1^ CON: basal diet with no additional fibre inclusion, LHC: basal diet including 1.5% low-hydration capacity insoluble fibre with fermentable fraction, MHC: basal diet including 1.5% medium-hydration capacity insoluble fibre with fermentable fraction, and HHC: basal diet including 1.5% high-hydration capacity insoluble fibre; ^2^
*n =* 8; ^3^ C1: CON vs. LHC, MHC and HHC; C2: LHC and MHC vs. HHC; C3: LHC vs. MHC.

**Table 5 animals-14-02612-t005:** Effect of dietary treatments on intestinal traits and caecal short-chain fatty acid concentration of piglets at 61 d.

	Dietary Treatment (DT) ^1^					*p*-Value ^3^			
Item	CON	LHC	MHC	HHC	SEM ^2^	DT	C1	C2	C3
pH of ileum	6.45	6.95	6.63	6.67	0.13	0.064	0.045	0.443	0.087
pH of colon	6.48	6.57	6.46	6.50	0.12	0.929	0.838	0.920	0.534
pH of caecum	5.83	5.92	5.66	5.88	0.12	0.467	0.912	0.550	0.146
Caecal weight, g	147	166	178	149	20.3	0.672	0.477	0.363	0.670
Caecal weight, % body weight	7.24	7.31	8.23	7.33	0.96	0.868	0.727	0.713	0.504
Caecal short chain fatty acids (SCFA), µmol/g									
Total SCFAs	239	246	209	235	14.4	0.297	0.590	0.664	0.076
Acetate	143	145	130	145	8.41	0.521	0.714	0.453	0.218
Propionate	63.1	64.0	51.1	59.7	4.33	0.160	0.339	0.695	0.040
Isobutyrate	0.72	0.69	0.76	0.70	0.15	0.990	0.972	0.913	0.757
Butyrate	28.0	32.5	24.3	25.9	3.13	0.303	0.914	0.522	0.080
Isovalerate	0.86	0.74	0.64	0.91	0.11	0.340	0.471	0.123	0.515
Valerate	3.26	3.82	2.47	3.21	0.79	0.690	0.917	0.943	0.237
Individual SCFAs, mol/100 mol									
Acetate	60.1	59.2	62.0	61.7	1.50	0.50	0.623	0.535	0.191
Propionate	26.4	25.8	24.5	25.2	0.86	0.42	0.213	0.967	0.269
Isobutyrate	0.30	0.30	0.36	0.32	0.070	0.925	0.794	0.887	0.544
Butyrate	11.5	12.9	11.7	10.9	0.90	0.487	0.733	0.237	0.345
Isovalerate	0.36	0.31	0.30	0.41	0.051	0.465	0.745	0.124	0.950
Valerate	1.33	1.50	1.20	1.41	0.34	0.936	0.923	0.894	0.538

^1^ CON: basal diet with no additional fibre inclusion, LHC: basal diet including 1.5% low-hydration capacity insoluble fibre with fermentable fraction, MHC: basal diet including 1.5% medium-hydration capacity insoluble fibre with fermentable fraction, and HHC: basal diet including 1.5% high-hydration capacity insoluble fibre; ^2^
*n* = 8; ^3^ C1: CON vs. LHC, MHC and HHC; C2: LHC and MHC vs. HHC; C3: LHC vs. MHC.

## Data Availability

Data are available upon request to the corresponding author.

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
