# Peer review of "Different Physiochemical Properties of Novel Fibre Sources in the Diet of Weaned Pigs Influence Animal Performance, Nutrient Digestibility, and Caecal Fermentation"

_animals, 2024, doi:10.3390/ani14172612_

Round 1
Reviewer 1 Report (New Reviewer)
Comments and Suggestions for Authors
This study aimed to investigate effect of dietary inclusion of fibre sources differing on fermentability and hydration capacity on performance, nutrient digestibility and caecal fermentation in post weaned piglets. Generally, this is an interesting study. However, fermentation capacity and hydration capactiy are two different characteristics of fibers. The high, medium, and low fermentation hydration capacity proposed by the author are incorrect in this study, and the design of the groups is completely wrong. The gas production rate represents the fermentation capacity, which can also be seen from the in vitro fermentation test. However, the gas production rate of the high fermentability and hydration capacity group is the lowest. The author needs to reorganize the content of the paper and submit a revised manuscript before considering whether to accept publication.
specific comments:
1. In this study, the authors used the treatment diets based on the inclusion of 1.5% fibre sources, however, no significant differences on most of parameters among groups were found, the dosage may be the reason for it. Why use 1.5%? 2. The authors found that addition of fiber had no effect on cecal SCFA, which is inconsistent with expectations. The dosage used may be one reason. In addition, the effect of fiber on different segments of the large intestine may be different.3. The authors should discuss the limitations of this study.
Author Response
Dear Reviewer
We greatly appreciate all your comments and suggestions. We believe that your contribution has improved the quality of the manuscript. Please see the attachment with all the answers to your comments. The changes are highlighted in yellow within the manuscript.
Bests regards

Reviewer 2 Report (New Reviewer)
Comments and Suggestions for Authors
Comments and Suggestions for Authors
The paper titled "Effect of dietary inclusion of fiber sources differing on fermentability and hydration capacity on performance, nutrient digestibility and cecal fermentation in post weaned piglets from 28 to 61 d of age.," authored by Agnieszka Rybicka et al., investigated the micronized fiber sources (FS) which differing in fermentability and hydration capacity (HC), on growth performance, faecal digestibility, and caecal fermentation in weaned piglets. Results of this study indicated LHC and MHC piglets showed an increase in daily growth and feed intake and tended to have a reduced FCR compared to HHC piglets. And fecal protein digestibility increased by 5% in the LHC and MHC groups compared with the HHC group at 42 days, significantly. These benefits may be attributed to LHC and MHC fibers enhancing intestinal and cecal environments, such as by increasing SCFA concentrations. These results are interesting and suggest that agricultural by-products could serve as an alternative fiber source in animal nutrition. The manuscript is well-written. The reviewer has only some minor concerns as follows:
1. In the introduction section, authors may describe the role of short chain fatty acids on gut microenvironment and growth performance. This may more reasonable the SCFAs analysis in this study.
2. Although LHC and MHC fibers were more fermented in vitro with cecal inocula than HHC fibers from 61-day-old piglets, the authors could discuss the potential benefits of these findings in the discussion section.
3. Lines 307 should be added “3.2” in title.
4. As the authors mentioned in the conclusion section, “the combination of insoluble fiber with a prebiotic fraction from …… provides advantages in performance and crude protein digestibility compared to lignocellulose…….” The authors may consider discussing the benefits of combining insoluble fiber in this concept in the discussion section.
Comments on the Quality of English LanguageNone
Author Response
Dear Reviewer
We greatly appreciate all your comments and suggestions. We believe that your contribution has improved the quality of the manuscript. Please see the attachment with all the answers to your comments. The changes are highlighted in yellow within the manuscript.
Bests regards

Reviewer 3 Report (New Reviewer)
Comments and Suggestions for Authors
Dear authors,
many thanks for this piece of work, which I read with interest and I found meritorious of being considered further.
While considering the topic of interest, I feel that some major flaws are present in the manuscript.
One of those regards the scientific literature authores referred to. Indeed, I found it unbalanced from a conceptual point of view, thus not introducing exhaustively the problem, nor commenting your results adequately. Likewise expressed in earlier comments, I felt that the reference is too much redundant on some aspects meanwhile neglecting many important others. Your introduction indeed raises the points which are not supported by the adequate description of the state of the art. On the other side, both introduction and discussion paragraphs are redundant on some other topics, specially the microbiome aspects. Which is in a way necessary and not inappropriate, but you titled your manuscript dealing with fibre and physical properties to be used in pig nutrition.
By the way, I would suggest the authors to shorten the title, which is definitely too long. My suggestion is "Different physico-chemical traits of micronized fibre in the diet of weaned pigs prove effective in modulating animal performance, nutrient digestibility and fermentation products". I hope authors would accept my suggestion. It sounds better, in my opinion.
Introduction:
I am not going to repeat myself, but please, improve the references to adequately describe the state of the art and the most updated literature. You may use the keywords physical form, dietary fibre, particle size, raw (which may be necessary to introduce your hypothesis on micronized), faeces quality. Some authors who investigated in the last decades such aspects Kamphues J., Zentek J., Cappai M.G., Papenbrock, Brouns F., Svihus B., etc... just as few examples. None of those investigations is reported, which laid down the use of dietary fibre, either treated or raw/native to have effects on post-weaned piglets, to control Salmonella for instance.
M&M:
L. 98: Please, prefer sex ratio: 50% to half females and half males.
L- 99: I am wondering if you reported the correct average body weight of piglets at 28th day of age. An average of 6.9 kg BW at 28 d appears slightly low to me for those breeds. Is this normal? Was it caused by litter size? Environmental temperature? Supplementation? Infections?
L. 129: Please, report in full the estimation of net energy (MJ/kg as is? Why not on DM basis?) below the Table 2. I agree with the list of ingredients as fed of course, but please, refer to all analytical value on DM in the diets please, including all nutrients and not only crude protein and fibre fractions. Those are necessary to estimate the energy content. Moreover, you used an AIA marker so please report Organic Matter. Thank you.
LL. 134-138: The feeding practice is not described, please add. Moreover, pigs were housed in boxes but how many boxes (and consequently pigs) belonged to one same treatment (I believe 8)? You report rooms of the plant where experimental boxes were placed, but this is not informative of the dietary treatments. Please, add daily intake and how those were determined and averaged. Please, report because this is missing in toto. You should be very sharp and exhaustive in describing this.
L. 136: Please, define Celite and refer to papers where this is used as digestibility marker for pigs.
L. 137: Please, prefer apparent total tract digestibility (ATTD) to fecal digestibility and add "of nutrients".
L. 154: "defection" a typo???? where feces collected from the ground???? Was there the possibility of mixing? This is one of the reason why the pen is the experimental unit and not the single pig.
L. 154-155: I have concerns on the fecal sampling method. It is not scientifically accurate. You collect feces differently from three animals in one pen where 6 pigs (why not 6/6????) are kept and may possess different performance and intakes and fecal outputs. This is not sound. Performance when mediated are referred to the whole pen, which represents the research unit, therefore not the single animal represents the valid research unit. This is the limit when large numbers of animals are used. Single animal performance cannot be representative unless a strict protocol is used and pooling method justified. Thus, the use in the statistical analysis/model of data from individual piglets is not sound when "alternatively" out of each pen. Indeed what comes out of each experimental unit (the pen) is the pooling of all animals of the pen and not the single piglet.
LL. 304-306: antibiotic treatments should have had an effect on your results. What about the clinical conditions and follow up? Veterinary assessment is missing to establish why authors decided all piglets would be kept within the experiment.
Comments on the Quality of English Language
Dear Editor,
the manuscript needs to be proofread.
Author Response
Dear Reviewer
We greatly appreciate all your comments and suggestions. We believe that your contribution has improved the quality of the manuscript. Please see the attachment with all the answers to your comments. The changes are highlighted in yellow within the manuscript.
Bests regards

Round 2
Reviewer 1 Report (New Reviewer)
Comments and Suggestions for Authors
No
Author Response
Dear Reviewer,
Thank you for the positive evaluation of our work.
Best regards
Reviewer 3 Report (New Reviewer)
Comments and Suggestions for Authors
Dear authors
Thank you for accepting most of suggestions. However, I feel that some requests of change which I would again recommend were discarded without sound consideration. Physical form and fermentation as well as GIT adaptation are of particular interest in this case. So please, look for physical form and morphological modification of ileum in weaning piglets.
Comments on the Quality of English LanguagePlease, let the manuscript proofread.
Author Response
Dear Reviewer,
In the manuscript attached you may observe all the changes highlighted in according to your suggestion. We also have marked in green those references which refers to issues you have pointed out.
In the current research we have not studied the impact of the feed form, since all the experimental diets have been prepared in pellet form. We have tried to fit the research of Papenbrock et al. (2005) in the introduction section, but finally we cannot see the connection of our work with that that studied the coarse grinding of main ingredients, with strong implications in particle size of the grain endosperm -and accordingly on starch particles-, but did not study any fibrous ingredient or fibrous feature.
We believe you will find well all the changes we have performed
Best regards
This manuscript is a resubmission of an earlier submission. The following is a list of the peer review reports and author responses from that submission.
Round 1
Reviewer 1 Report
Comments and Suggestions for Authors
Dear editor,
I have reviewed the manuscript entitled: “Effect of dietary supplementation of fiber sources differing on fermentability and hydration capacity on performance, nutrient digestibility and cecal fermentation in post weaned piglets from 28 to 61 d of age.”. In general, I found the manuscript interesting, well written, and informative.
I have made some comments in the file attached to this review. However, in my opinion, discussion is the manuscript's section that requires more significant improvements. In some cases, it is difficult to follows and understand. My central suggestion to the authors is to improve the wording of this section to present their ideas in a better flow and reduce the number of lines in the paragraphs. In addition, in several parts, authors compared your data with literature findings, but no inferences from those comparisons are provided.
I hope to contribute to improving the quality of this manuscript with this review.
Regards,

Comments on the Quality of English LanguageIn my opinion, this paper needs significant improvements in the wording of discussion section. Check wording to improve the flow of ideas.
Author Response
Dear Reviewer,
We greatly appreciate all your recommendations. We deeply believe that all your comments have improved the quality of this manuscript.
Per your suggestion, we have changed the treatment abbreviation in the manuscript to make it more intuitive. We hope you find the new abbreviation (CON, LHC, MHC and HHC) good enough. Also, the Table 1 and Table 2 have been interchanged. However, due to the large size of Table 2, it should be put on the whole page. According to your recommendation, several citations have included the first author’s name, and some sections of the discussion have been split for better understanding. We also applied your suggestion in the Tables and Figures
The manuscript will be English-reviewed once all the reviewers accept the changes proposed.
Please see the attachment, where you will find the list of the corrections made. In the manuscript, we have marked in blue color all the changes you suggested.
Best regards,

Reviewer 2 Report
Comments and Suggestions for Authors
The subject of paper “Effect of dietary supplementation of fiber sources differing on fermentability and hydration capacity on performance, nutrient digestibility and cecal fermentation in post weaned piglets from 28 to 61 d of age” falls within the scope of the journal. This research work tackles the inclusion of different fibre sources in pig diets, which is an aspect so far insufficiently studied in monogastric animals. Hence, the study could be interesting and useful in order to broaden the knowledge on pigs’ intestinal utilisation of fibre. However, several unavoidable improvements need to be done in the paper before it can be accepted for publication. Failure to complete these improvements would justify the reject of this manuscript.
Major flaws of the manuscript submitted:
1- Authors say they are testing different fibre sources that differ on fermentability and hydration capacity. However, authors do not properly characterise the fibre sources they are including in the diets. What are the differences in fermentability? How has fermentability of fibrous ingredients been measured? Where are these differences shown in the paper? The proportions of the different fibrous ingredients within the mixture are not provided. How much of almond shell? How much of olive kernel? Did grape pomace originate from fermented or unfermented grapes? The latter is very important to be specified. This can largely influence the fermentability of this ingredient in the intestine of pigs. Furthermore, how much of chicory root is contained in the fibrous mixtures? With the current version of this manuscript, readers would not be able to know which fibre sources and in which doses are being included in each experimental diet.
2- This research work is very poor in results. This study deals with the assessment of the intestinal utilisation of different fibrous ingredients in piglets. Hence, interesting dependent variables that should have be measured are the digestibility coefficients of soluble fibre and that of neutral-detergent fibre, the gut morphology (villus length, crypt depth, mucin content, etc…), the caecal counts of cellulolytic bacteria, etc… I wonder whether the paper can be accepted for publication without providing these results…. At least, the results on the digestibility of energy are unavoidable.
3- Line 107: Authors mention that fibrous ingredients were finely-ground. How was grinding performed? What was the size targeted?
4- This research is primarily based on the fermentation of fibrous ingredients by the caecal microbiota. However, no characterisation at all of the microbiota is provided. As aforementioned, the population of cellulolytic bacteria in the caecum should have been quantitatively assessed. Since this seems not to have been performed in the current study, a thorough bibliographic review of the cellulolytic microflora present in the pig caecum should be done and included in the discussion of this paper in order to support the statements done by the authors. Otherwise, authors are not allowed to do most of the statements so far contained in the discussion of this manuscript.
5- This manuscript has to be proofread by someone with a high knowledge of English. Numerous grammar, spelling and conjugation mistakes can be found all throughout the manuscript. These mistakes can be easily corrected by someone proficient in English. Moreover, writing of the text can also be improved in order to ease the reading of the document.
6- British spelling is to be used in MDPI journals. Please modify the whole manuscript accordingly.
Additional comments:
Title: you should say “dietary inclusion”, not “dietary supplementation”. You are including your fibre mixture as an ingredient within the diet. You are not providing a dietary supplement like an enzyme or a synthetic aminoacid.
Line 15: remove comma before “remain”
Lines 18 and 19: “their hardness”, “their physicochemical”, “their fermentability”
Line 74: “their hardness and size but when properly milled, they can be…”
Line 76: “which modifies the physical…”
Line 78: “increase” instead of “increased” because you are saying that “this process may increase”
Line 98: Pigs were weighed, not weighted.
Line 100: Explain how the different treatments were distributed in the four rooms.
Line 103: Provide details about the light programme used in the rooms.
Line 110: with both mixtures including chicory root….
Line 110: at which dose was chicory root included?
Line 124: Piglets were randomly assigned. Without taking into account the sex of the animals?
Line 124: “to one of the 4 dietary treatments”
Line 124-125: If the homogeneity in the initial body weight was taking into consideration when allocating the piglets to the four different dietary treatments, why was it then necessary to include weaning body weight as a covariate in the statistical model?
Lines 125-126: the meaning of this sentence is not clear: “A total of 96 males and females were allocated separately”. It could be understood that piglets were housed individually. However, it is mentioned that piglets were grouped (six in each pen).
Lines 127-128: “pen feed consumption” instead of “pen feed consume disappearance”
Table 2 should be placed before current Table 1. Fibre sources should be characterised before providing the composition of the experimental diets.
Table 1: The calculated nutrient composition of the diets is not acceptable. Authors should provide the analysed nutrient composition of diets, at least for crude protein, neutral detergent fibre, acid detergent fibre, lignin and soluble fibre. They are also requested to provide the analysed digestible energy content of diets.
Footnote of Table 1: “Provided”, not “Provied”
Tables should be self-understandable. In Table 1, authors have to specify what the ingredient called “Fibre source” is.
Table 2: According to the title, this table shows the composition of the fibre sources. However, columns show the composition of the experimental diets, not that of the fibre sources. A thorough characterization (doses of the different fibrous ingredients and content in the different fibrous fractions) is compulsory. If not provided, this paper cannot be accepted for publication.
Line 145: “pre-stunned”. Please use the correct form of the verb.
How were the piglets pre-stunned? And then how were they slaughtered?
Line 148: the whole caecum or only the caecal contents?
Line 148: Again, “weighted”. It should be “weighed”.
Line 153: Again, “weighted”. It should be “weighed”.
According to lines 495-496, “All authors have read and agreed to the published version of the manuscript”. I’m surprised that none of them noticed the frequent grammar errors in this manuscript.
Line 156: was the microbiota characterised?
Lines 160-162: Please rephrase this sentence. So far it is not clear. If one piglet is slaughtered per pen, this means that eight piglets were slaughtered per treatment. Then, the caecal contents of two piglets were pooled, so you got four pools of caecal content per treatment. How come you finally obtained twelve inocula?
Line 180: strike “in” after “Insoluble and soluble fibre were determined”
Line 246: How was the Total Dietary Fibre concentration determined? This has not been explained nor have these data been shown in the tables.
Line 254: “were properly designed”
Lines 257 and 261: “on average”
Table 3: since you are stating that feed intake and weight gain were measured on a daily basis (ADFI and ADG), the correct unit is merely “g”, not “g/d”
Table 4: the digestibility of energy should also be provided. This result is unavoidable. Moreover, the digestibility coefficients of soluble fibre and neutral-detergent fibre would also have been interesting, since this work deals with the intestinal utilisation of the different fibre sources.
Line 400: you cannot assert that “Dietary treatment showed no effects on caecal morphology” because you have not measured any trait pertaining to caecal morphology.
Line 431: “in agreement with”
Line 434: “their fermentable fraction”
Lines 434-436: the intestinal fermentative capacity largely differs between pigs and chickens. Results are not comparable at all. Remove these comparisons.
Line 455: “which is consistent with our results”
Lines 466-469: you cannot surmise that the fibre sources used in this study might have an impact on microbiota profile since you did not characterise caecal microbiota at all. A thorough bibliographic review of the cellulolytic microflora present in the pig caecum should be done and included in the discussion of this paper in order to support the statements done by the authors.
Line 474: I’m surprised to read “the fibre had already been fermented in the previous part of the intestine”. Where? As largely reported in published research works, caecum is known to be the main place of fibre fermentation in swine.
Line 488: you did not measure the fructan content in the chicory root or if measured, this concentration has not been provided in the manuscript. As long as the concentration of fructans is not provided, you cannot state that "the inclusion of fructans from chicory root may increase the faecal digestibility of crude protein….".
The conclusions of this manuscript need to be rewritten from scratch. At present, conclusions seem to be a summary of the results. Instead, the conclusion section of the manuscript should distinctly express the primary findings from the study. These findings should be directly tied to the study’s objectives, as stated in the introduction or derived from the tested hypothesis.
Comments on the Quality of English Language
This manuscript has to be proofread by someone with a high knowledge of English. Numerous grammar, spelling and conjugation mistakes can be found all throughout the manuscript. These mistakes can be easily corrected by someone proficient in English. Moreover, writing of the text can also be improved in order to ease the reading of the document. Some of the grammar and conjugation mistakes are listed in the additional comments.
Author Response
Dear Reviewer,
We greatly appreciate all your recommendations. We deeply believe that all your comments have improved the quality of this manuscript.
However, regarding the proposals related to the analysis of new different parameters, we would like to point out that this research had a limited budget, and a number of the traits analyzed were selected according to the main objectives planned for this project. For future trials, we will try to include the variables that you gently suggested.
Indeed, we would like to thank you very much for all your comments, and we tried to cover all of them.
Please, see the attachment where you will find the list of the corrections made.
Best regards,

Reviewer 3 Report
Comments and Suggestions for Authors
This research had a double purpose. Firstly, to evaluate in piglets feeding the use of novel fiber sources such as almond shell, olive kernel or nutshell as very finely ground fiber sources, which commonly are not suitable as dietary ingredients due to its hardness and particle size. Secondly, to assess the impact of its physicochemical properties such as its fermentability and hydration properties on piglets’ performance, compared with lignocellulose that is the most commonly used source of fiber. Overall, these results point out the importance of the physicochemical characteristics of fiber sources, suggesting that a combination of insoluble fiber with a prebiotic fermentable fraction, having medium hydration properties, may provide good results on performance in weaned piglets. These studies were done with procedures and techniques appropriate for these types of studies. However, there are some concerns that need to be clarified.
1.Are experiments well documented? Is there information on positive and/or negative controls, and are the number of replicates and/or sample sizes provided?
2.Please check and modify the format of some references in the article.
3.Please check and modify the format of some references in the article.
4.Does the reference list cover the relevant literature adequately and in an unbiased manner?
5.Follow the journal’s style for writing the headings, and sequence of headings, etc. please
6.Are the methods sufficiently documented to allow replication studies?
7.The conclusion is not very clear, please elaborate on it.
8.Please review the language and format of the text and revise it accordingly.
Comments on the Quality of English Language
no comments
Author Response
Dear Reviewer,
We greatly appreciate the suggestions and changes recommended. Below you will find the corrections. The manuscript will be sent for the English level revision.
Please see the attachment, where you will find all the answers for your questions.
Best regards,

Round 2
Reviewer 1 Report
Comments and Suggestions for Authors
Dear authors,
I reviewed again the paper, and it was much improved. The authors carefully revised all my comments, and most were applied to the manuscript. Hence, in my opinion, the manuscript can be accepted for publication in its current form.
Regards,
Reviewer
Comments on the Quality of English LanguageI suggest authors made the English revision of the manuscript prior the submission of revised version. Without the English revision is difficult to evaluate the quality of the manuscript as as whole.
Author Response
Dear Reviewer,
We greatly appreciate all the recommendations we have received from you. The manuscript will be English reviewed once we have the final version, with all the changes proposed.
Best regards,
Reviewer 2 Report
Comments and Suggestions for Authors
It must be acknowledged that authors have made an effort in order to improve the quality of their manuscript. Yet, many issues are to be solved before this paper becomes eligible for publication.
To begin with, the subsequent comments already included in the first step of this revision process have not been taken into account:
Lines 73-75: “their hardness and size but when properly milled, they can be…”
Line 76: “which modifies the physical…”. You are talking about a novel technology, so it should be “modifies” instead of “modify”.
Line 100: Explain how the different treatments were distributed across the four rooms.
Table 1: the spelling “fibre” should be used in the title and in the heading
Table 1: the spelling “characterised” should be used
Lines 168 and 169: the spelling “caecum” should be used
Lines 188-189: this request has not been addressed: Please rephrase this sentence. So far it is not clear. If one piglet is slaughtered per pen, this means that eight piglets were slaughtered per treatment. Then, the caecal contents of two piglets were pooled, so you got four pools of caecal content per treatment. How come you finally obtained twelve inocula?
Table 4: The digestibility coefficients of soluble fibre and neutral-detergent fibre should be provided, since this work deals with the intestinal utilisation of the different fibre sources.
Lines 486-489: authors cannot surmise that the fibre sources used in this study might have an impact on microbiota profile since they did not characterise caecal microbiota at all. A thorough bibliographic review of the cellulolytic microflora present in the pig caecum should be done and included in the discussion of this paper in order to support the statements done by the authors.
Lines 494-495: Still, it has not been explained where in the intestine fibre may have been fermented. Previous comment was: Where? As largely reported in published research works, caecum is known to be the main place of fibre fermentation in swine.
Furthermore:
Lines 23-25: Rephrase as “The effect of including in the diet micronized fibre sources (FS), differing on fermentability and hydration capacity (HC), on growth performance, faecal digestibility and caecal fermentation were investigated in 192 piglets (Landrace x Duroc) aged 28 d at the beginning of the trial”
Line 76: “micrometre” instead of “micro meter”
Line 101: Add “daily” in “(40 lux during 8h daily)”
Table 2: if the analysed digestible energy content of diets is not available, at least the calculated digestible energy content should be provided for the experimental diets. This would enable to confirm whether diets were isocaloric, as asserted by the authors in the text.
Furthermore, it must be specified in the Table that “Net energy (MJ/kg)” is estimated and in the text, it should be explained how the dietary net energy content was estimated.
Line 166: the word “gun” is missing in “with a captive-bolt gun”.
Line 227: “Berrocoso et al. and Priester et al.”
Line 235: a blank is needed before subheading 2.8.3
Line 289: this is Table 3
Table 3: FCR is an adimensional parameter. It has no units.
Line 305: this is Table 4
Lines 452-454: I keep on considering that these species are not comparable. The intestinal fermentative capacity largely differs between pigs and chickens. These comparisons are not scientifically valid and should be deleted.
Line 493: the spelling “caecum” should be used.
Comments on the Quality of English Language
Improvements have been performed from the previous version of the manuscript but still some mistakes are present and remain to be corrected.
Author Response
Dear Reviewer,
We kindly appreciate all your suggestions that make the manuscript better. Please see the attachment with all the answers.
Best regards,
